# Sleeping Disorders in Healthy Individuals with Different Dietary Patterns and BMI, Questionnaire Assessment

**DOI:** 10.3390/ijerph182312285

**Published:** 2021-11-23

**Authors:** Magda Piekarska, Martyna Pszczółka, Damian Parol, Paweł Szewczyk, Daniel Śliż, Artur Mamcarz

**Affiliations:** 1Lifestyle Medicine Student Interest Club, 3rd Department of Internal Diseases and Cardiology, Medical University of Warsaw, 04-749 Warsaw, Poland; mpszczolka@hotmail.com; 23rd Department of Internal Diseases and Cardiology, Medical University of Warsaw, 04-749 Warsaw, Poland; parol.damian@gmail.com (D.P.); sliz.daniel@gmail.com (D.Ś.); artur.mamcarz@wum.edu.pl (A.M.); 3Division of Dietetics, Department and Clinic of Gastroenterology and Hepatology, Wroclaw Medical University, 50-556 Wroclaw, Poland; dietetyk.pawel@gmail.com; 4Public Health School Postgraduate Medical Education Center, 01-813 Warsaw, Poland

**Keywords:** sleep, lifestyle medicine, vegetarian, nutrition, plant based, insomnia, prevention, sleep quality, Poland, BMI, sleepiness

## Abstract

The COVID-19 pandemic and abiding restrictions have affected every life domain. Sleep disturbances are a major health issue that is linked with a higher prevalence of metabolic syndrome, obesity, and psychological burdens. Research of sleeping disorders among vegetarian and non-vegetarian subpopulations is limited. The aim of the study was to assess the prevalence of sleeping disorders during the COVID-19 pandemic among people with different dietary patterns. Using a web-based cross-sectional survey, data were collected from 1987 people. A total of 1956 respondents met all study conditions. The questionnaire consisted of sociodemographic information, assessment of dietary habits, and assessment of the prevalence of insomnia and sleepiness, based on the Athens Insomnia Scale (AIS) and Epworth Sleepiness Scale (ESS). A total of 36.04% (*n* = 705) respondents declared that they noticed a change in the quality of their sleep during the last year. According to AIS and ESS, non-vegetarians suffered from insomnia or sleepiness more often than vegetarians. Insomnia and sleepiness were also more prevalent among those respondents who declared consumption of fruit and vegetables less often than once a day compared with those who consumed fruit and vegetables daily. Respondents with BMI within the recommended limit (18.5–24.99) suffered from insomnia less often when compared with underweight (BMI < 18.5) or obese (BMI ≥ 25) respondents. Those results may be useful for public health workers and medical professionals in terms of establishing new instruments that help treat sleeping disorders.

## 1. Introduction

At the beginning of 2020, the World Health Organisation announced COVID-19 as a Public Health Emergency of International Concern [1]. Throughout the last year, governments all across the world implemented various restrictions to slow the spread of the virus. One of the most important preventative measures which were implemented was social distancing [2]. Physical distancing and quarantines greatly changed our lifestyles, which contributed to increasing levels of stress and anxiety [3]. What is more, previous epidemic situations showed that the number of people whose mental health was affected may be greater than the number of people infected by the disease itself [4]. Therefore, sleep patterns should be an important subject of interviewing a patient, and more awareness of the problem among workers is needed. Guidelines for preventing the psychological impacts of quarantines [3] or preventing suicide [5] may be very helpful in this unusual situation. 

This study does not intend to examine the influence of the COVID-19 pandemic on sleep quality. However, it is important to highlight the fact that the study was conducted during the COVID-19 pandemic, as it could have interfered with the results.

Sleep is a process that is instrumental in maintaining psychological, emotional, and physical health. Growing data indicate that poor sleep is associated with a wide range of adverse health effects. Sleep deficiency may be related to such consequences as obesity, type 2 diabetes, hypertension, or cardiovascular disease [6,7,8,9,10]. The Sleep Heart Study found that subjects with insomnia or poor sleep with short sleep were at 29% higher risk of cardiovascular disease [11]. Sleep disturbances may also have an impact on premature mortality [8]. Furthermore, poor sleep is linked with social difficulties, such as adverse performance at school and in the labor market. It is also worth to mention breathing-related sleep disorders which include habitual snoring, upper airway resistance syndrome (UARS), and obstructive sleep apnea (OSA) [12]. If underdiagnosed or untreated, it may lead to serious consequences, such as cardiovascular diseases, type II diabetes, and depression [13]. Therefore, it is pivotal to screen for these conditions using proper diagnostic tools [14]. Due to the above-mentioned consequences, sleep problems represent a major public health concern.

Numerous research papers indicate an association between diet and sleep quality, especially its duration. Insufficient sleep is related to unhealthy diet choices during the day [15,16,17,18]. According to research, people whose sleep is shorter tend to consume more calories [15,16,19] compared with people whose sleep time is sufficient. It is also observed that shorter sleep time is linked with a higher intake of fat and carbohydrates during the day [15,16]. However, available research does not show a clear causality: does poor sleep affect diet choices, or does an unhealthy diet cause poor sleep? Furthermore, there is little research analyzing the association between a vegetarian diet and sleep quality. 

Another study [20] showed that higher levels of meat consumption was correlated with poor sleep quality, and a decrease in sleep duration. Therefore, the study hypothesis was that sleep quality may vary between people with different dietary patterns, which may translate to differences in the prevalence of sleeping disorders. The aim of this study was to assess the prevalence of sleeping disorders, such as insomnia and sleepiness among vegetarian and non-vegetarian subpopulations. Another intention of the study was to examine its associations with dietary and lifestyle variables, such as different meat consumption levels and different fruit and vegetable consumption levels, among other factors.

## 2. Materials and Methods

### 2.1. Study Design and Participants

To collect data, a web-based cross-sectional survey was performed. The questionnaire was designed using the survey administration application Google Forms, and it was self-administered. Anonymity and confidentiality were ensured. Filling in this questionnaire was completely voluntary. A statement about the aims of the study and the declaration of anonymity and confidentiality were included in the form. Filling in the questionnaire and clicking the “send” button was synonymous with informed consent to participate in the study.

The questionnaire consisted of four main sections: socio-demographic questions, assessment of factors that may influence sleep quality, dietary assessment, and assessment of the prevalence of sleeping disorders such as insomnia and sleepiness. The questionnaire form is available in Appendix A. The questionnaire was distributed via Polish Facebook groups and profiles. The participants were random people from Poland who were members of social media groups. The only inclusion criteria were proper filling out of the survey and clicking the “send” button as a sign of consent to participate in the study.

### 2.2. Data Collection

The questionnaire was available online and could be filled in by the participants anonymously from 5 February 2021 to 23 February 2021. The data were collected from 1987 people. A total of 31 questionnaires were excluded from the analysis due to inaccurate answers (e.g., incorrect weight or height, incorrect answers in the dietary section). Finally, 1956 participants met all study conditions and were included in the analysis.

### 2.3. Measures

#### 2.3.1. Demographic Information and Declaration of Diet

Demographic variables included gender (male or female), age, weight, height, and diet. Diet patterns included vegetarian, vegan, or including meat. Vegetarians were defined as people who excluded flesh foods (such as meat, poultry, wild game, seafood, and their products); vegans were defined as people who excluded all animal products from their diet (flesh foods, eggs, dairy products) [21]. Later, when referring to “vegetarians”, both vegetarians and vegans were included in that group. This part of the survey also included a question, “How long have you been vegetarian?” Due to the short intervention time, respondents who declared that they had been vegetarians for no longer than 6 months were classified as “non-vegetarians”.

#### 2.3.2. Assessment of Factors That May Influence Sleep Quality

This section consisted of questions about factors other than diet which may exert an impact on sleep quality. These variables included smoking, shift work, psychiatric disorders [22], sleep apnea, and intake of medicines [23] which may influence sleep. The questionnaire also included questions about behaviours performed before going to bed, such as using electronic devices emitting blue light, drinking coffee or alcohol, and exercising. 

#### 2.3.3. Assessment of Dietary Habits

For the assessment of dietary habits, the Polish questionnaire KomPAN [24] was used. Due to the limitations of the online survey, we used a minimal set of questions (which included 29 questions) indicated by the authors of this questionnaire in the scoring protocol [24]. This short form of the KomPAN questionnaire was validated by its authors [24]. In this survey, respondents were asked to report on the frequency of consumption of particular foods during the last year. This section included three verifying questions. A total of 26 participants were excluded from the analysis on their basis. The komPAN questionnaire is available in Appendix A. 

#### 2.3.4. Assessment of the Prevalence of Sleeping Disorders

In this part, the Athens Insomnia Scale (AIS) [25] was used to assess the prevalence of insomnia, and the Epworth Sleepiness Scale (ESS) [26] to assess the prevalence of sleepiness. 

AIS consists of eight items: difficulty with sleep induction, awakenings during the night, early-morning awakening, total sleep time and overall quality of sleep, problems with their sense of well-being, functioning, and sleepiness during the day. Each item can be rated from 0 (“no problem at all”) to 3 (“very serious problem”). A score of 6 points was established to be an optimum cut-off for insomnia [25].

The ESS asks respondents to rate, from 0 to 3, their usual chances of having fallen asleep while being engaged in eight different activities. A score greater than 10 points indicates sleepiness [26]. 

### 2.4. Statistical Analysis

Statistical analysis was conducted in Statistica (TIBCO Software Inc. (2017). Statistica (data analysis software system), version 13. http://statistica.io.(Accessed on: 21 November 2021). Basic statistical calculations were made (mean, standard deviation). A chi-squared test was conducted to correlate insomnia and sleepiness with other factors, such as a vegetarian or non-vegetarian diet, level of meat consumption, level of fruit and vegetable consumption, and BMI. The findings were considered statistically significant if the *p*-value was less than 0.05.

## 3. Results

### 3.1. Demographic Characteristics

The characteristics of participants were shown in Table 1. Of the 1956 samples analyzed, 1649 (84.3%) were females. The mean age of the participants was 30 years (SD = 8.37). Among the examined population, 747 (38.2%) of participants were vegetarians and 1209 (61.8%) of participants were non-vegetarians.

### 3.2. Prevalence of Sleeping Disorders among Respondents

The prevalence of insomnia and sleepiness is shown in Table 2, Table 3 and Table 4.

A total of 36.0% (*n* = 705) respondents declared that they noticed a change in the quality of their sleep during the last year (February 2020–February 2021). According to AIS and ESS, 56.9% (*n* = 1112) respondents suffered from sleeping disorders; 49.1% (*n* = 961) respondents suffered from insomnia; and 23.0% (*n* = 451) suffered from sleepiness. Females suffered from insomnia or sleepiness more often than males: 50.2% females vs. 43.2% males suffered from insomnia (*p* < 0.05, as shown in Table 2), and 24.9% females vs. 13.4% males suffered from sleepiness (*p* < 0.05, as shown in Table 2). Among females, vegetarians suffered from sleeping disorders less often than non-vegetarians: 46.6% vegetarian females vs. 52.7% non-vegetarian females suffered from insomnia; 21.3% vegetarian females vs. 27.3% non-vegetarian females suffered from sleepiness (Table 3). When males were taken into consideration, such associations were not statistically significant (as shown in Table 4). 

### 3.3. Prevalence of Sleeping Disorders Stratified by Diet Variables

The prevalence of insomnia and sleepiness stratified by diet variables is shown in Table 4.

According to AIS and ESS, non-vegetarians suffered from insomnia or sleepiness more often than vegetarians; 51.4% (*n* = 621) non-vegetarians vs. 45.5% (*n* = 340) vegetarians suffered from insomnia (*p* < 0.05); and 25.0% (*n* = 302) non-vegetarians vs. 20.0% (*n* = 149) vegetarians suffered from sleepiness (*p* < 0.05). Insomnia and sleepiness were also more prevalent among those respondents who declared consumption of fruit and vegetables less often than once a day; 55.4% (*n* = 133) respondents who consume fruit and vegetables less often than once a day vs. 48.2% (*n* = 828) respondents who consume fruit and vegetables at least once a day suffered from insomnia (*p* < 0.05); 30.0% (*n* = 72) respondents who consume fruit and vegetables less often than once a day vs. 22.1% (*n* = 379) respondents who consume fruit and vegetables at least once a day suffered from sleepiness (*p* < 0.05). Associations between a higher prevalence of sleeping disorders and more frequent use of alcohol, and drinking coffee 5 or fewer hours before going to bed were not statistically significant.

### 3.4. Prevalence of Sleeping Disorders Stratified by Lifestyle Variables

Smokers suffered from insomnia statistically significantly more often than non-smokers: 58.0% (*n* = 141) vs. 47.9% (*n* = 820), respectively (*p* < 0.05; as shown in Table 4). Using electronic devices 1 h before going to bed everyday was associated with a higher prevalence of insomnia: 50.8% (*n* = 873) vs. 37.3% (*n* = 88), (*p* < 0.05; as shown in Table 4).

### 3.5. Prevalence of Sleeping Disorders Stratified by BMI

The prevalence of insomnia and sleepiness among BMI groups is presented in Table 4.

Respondents whose BMI was within the recommended range (18.5–24.99) suffered from insomnia less often when compared with underweight (BMI < 18.5) or overweight (BMI > 24.99) respondents: 46.7%, 53.6%, and 55.1%, respectively (*p* < 0.05). Nonetheless, this association was not statistically significant for sleepiness. 

## 4. Discussion

The study was conducted to assess the sleep quality of Polish people and its associations with different dietary patterns during the COVID-19 pandemic. The analysis was focused on comparing vegetarian and non-vegetarian subpopulations. 

In the present research, we established that more than 1/3 of respondents noticed a change in the quality of their sleep during the last year (February 2020–February 2021). Other studies conducted during the COVID-19 pandemic report similar results [27,28]. One study shows an increase in bedtime hour, sleep latency, and wake-up time before and during the COVID-19 crisis [27]. In a few studies, worsening sleep quality was also observed as causing the increase of the total score in the Pittsburgh Sleep Quality Index (PSQI) [27,28]. Papazisis et al. conducted a study on the impact of the quality of one’s sleep and lifestyle during the second lockdown [29]. Compared to our study in which 36.0% of respondents noticed a change in sleep quality, in the Greek study, this number reached 43.5% (out of which 39.5% of respondents answered that the quality of their sleep improved and 60.5% of respondents answered that the quality of their sleep worsened). However, the study did not focus on sleep disturbances in detail.

Our study shows a high prevalence of sleeping disorders, especially insomnia among the examined population—the insomnia rate in our study reached 49.1%. In the PaLS study [30], which was also conducted in the COVID-19 period in Poland, 58.13% of the respondents were suffering from insomnia (measured with the Athens Insomnia Scale). The difference of 9 percentage points may be caused by the fact that the PaLS study was conducted among students; meanwhile, our study group demographic was more varied. In a study conducted in Italy [28], the rate of poor sleep quality (defined as a score higher than 5 points in PSQI) was 52.4%—a result similar to the rate of insomnia (defined as a score higher than 5 points in AIS) observed in our study. What is interesting is that a study from China reports that only 18.2% of respondents had poor sleep quality during the COVID-19 pandemic; however, the cut-off for poor sleep in this research was set at a score greater than 7 points in PSQI [31].

In our study, females suffered from sleeping disorders more often than males. Similar conclusions have been drawn in the literature [32]. In the PaLS study [30], which was also conducted in Poland, females more often suffered from insomnia. However, in a recent study from China, such an association was not observed [31]. Studies show that sleeping disorders are more common among women than men, with the exception of symptomatic obstructive sleep apnea, which is twice as common in men than women over the age of 50 years (4% and 2%, respectively) [33]. Obstructive sleep apnea (OSA) has been shown to be strongly associated with lower serum testosterone levels in men [34,35,36]. The association between low testosterone levels and OSA is bidirectional. Lower testosterone levels have been shown to worsen the efficiency of sleep, to increase the number of nighttime awakenings, and to reduce deep sleep time [37].

Vegetarians were found to suffer from insomnia and sleepiness less often than non-vegetarians. This association was also statistically significant when only comparing females. Moreover, consuming fruit and vegetables at least once a day was associated with a lower prevalence of sleeping disorders when compared with consuming fruit and vegetables less often than once a day. 

Numerous research papers indicate an association between diet and sleep quality. Sleep deficiency is related to poor diet choices during the day [15,16,17,18]. People whose sleep is shorter tend to consume more calories [15,16,19] and eat more unhealthy snacks [15,18] when compared with those whose sleep duration is sufficient. However, the causality of these relations is unclear. It is not known whether short sleep time causes poor diet choices, or whether a poor diet has an impact on sleep. 

There is little research regarding the association between a vegetarian diet and sleep quality. One research paper showed that among teenagers from Sicily who adhered to the Mediterranean diet, a higher consumption of fruit and vegetables was positively associated with sleep quality [38]. A similar association was found in our study. Another study from Japan showed that a high intake of vegetables and fish was related to better sleep quality [39]. Other foods that were beneficial for sleep mentioned in this research were mushrooms, potatoes, seaweed, soy, and eggs. High consumption of these products was associated with a lower risk of experiencing troubles with falling asleep. A cross-sectional study of women aged between 20–75 showed that a higher intake of plant protein is associated with better sleep quality and a lower prevalence of insomnia [40].

Some studies point out the possible explanations for the association between a plant-based diet and better sleep quality. One of the possible reasons for that is high isoflavone content in plant protein. A few studies [41,42] have shown that a high intake of that element is related to a lower risk of excessively long sleep (longer than 9 h) and a lower risk of falling asleep during the day among women. Another substance that may exert a positive impact on sleep is tryptophan [43]. This amino acid is a precursor of melatonin and serotonin, which contribute to sleep regulation [44,45,46]. What is more, reduced melatonin secretion may be involved in the mechanism of insomnia [47]. A study [48] showed that consuming cereals fortified with tryptophan before going to sleep caused prolonged sleep duration and enhanced sleep quality compared with consuming cereals not fortified with this amino acid, or with not consuming cereals at all. Plant-based products are rich in tryptophan, which may be the explanation for their positive impact on sleep.

Another factor that was associated with poor sleep in our research was smoking. Tobacco users suffered from insomnia more often than non-tobacco users. Other studies show similar results [49,50]. According to the research from China [49], cigarette smoking was associated with numerous sleep disturbances: subjective sleep quality, sleep latency, and sleep duration. According to the data derived from the SHADES study [50], conducted among Pennsylvania inhabitants aged 22–60, smokers had a significantly higher risk of insomnia (assessed on the basis of the Insomnia Severity index) than non-smokers (OR = 2.5). We came to a similar conclusion with cigarette-smoking respondents in our study who suffered from insomnia significantly more often than those who denied smoking, 58.0% vs. 47.9%, respectively (*p* < 0.05).

What is worth noticing is the fact that the vast majority (up to 87.9%) of respondents declared that they used electronic devices 1 h before going to bed every single day. These respondents also experienced insomnia more often than people who denied using electronic devices before going to bed every day. A study from Italy pointed out that the frequency of using digital media 2 h before going to bed increased during the pandemic restrictions [28]. Findings from another study indicated that using digital media at bedtime contributed to poorer or disrupted sleep [51].

Insomnia was also associated with BMI. Respondents with a BMI within the recommended range suffered from insomnia less often than underweight or overweight respondents. There are studies observing similar results. Papazisis et al. observed the influence of BMI on sleep. The difference is that their main criterion was sleep duration; meanwhile, in our study, sleepiness and insomnia were analyzed as defined above. In the Greek study, overweight and obese participants slept less (−0.44 h and −0.66 h, respectively) than normal weight participants; meanwhile, in our study, the aforementioned sleep disturbances were more common among respondents with BMI < 18.5 (underweight) and BMI > 24.99 (overweight, obese). Since the duration of sleep is one of the factors that define insomnia, a conclusion can be made that the results of the two studies are consistent with each other. Meta-analyses and systematic reviews observed that shorter sleep is associated with a higher BMI [52,53,54,55]. What is interesting is the bidirectionality of that association. However, this research does not answer the question on whether short sleep duration causes a change in BMI or a higher BMI results in a change in sleep duration.

When interpreting the results of this study, some limitations ought to be taken into consideration. The data were collected through a cross-sectional online-based questionnaire. Therefore, it is difficult to make a causal interference. What is more, the possibility of selection bias should be taken into account, as it is possible that people who completed our form were interested in a healthy lifestyle. Moreover, this study relied on self-reported questionnaires and did not include objective measures, such as polysomnography. It is also crucial to be cognizant of possible dietary measurement errors [56]. Due to the socio-cultural pressure to follow a “correct” dietary pattern, respondents whose intake of supposedly healthy food is low may be tempted to report their intake of these products as higher than it really is. Furthermore, the intake of unhealthy food may be underreported. To sum up, the fact that the questionnaire was web-based and self-reported could have negatively influenced data quality. Another limitation of our study is the fact that it is based on univariate analysis.

## 5. Conclusions

These findings suggest that the pandemic and abiding restrictions impacted people’s sleep quality. The rate of insomnia was high among the examined population. Vegetarians suffered from insomnia and sleepiness less often than non-vegetarians. What is more, higher fruit and vegetable consumption was associated with a lower insomnia rate. However, further research is needed to establish the causality of these associations. Research findings relating to the correlation of diet and sleep quality may be useful for creating guidelines for enhancing sleep quality.

## Figures and Tables

**Table 1 ijerph-18-12285-t001:** Sociodemographic characteristics.

Category	Variables	Number	Percentage
Gender	Male	307	15.7
	Female	1649	84.3
Diet	Vegetarian	747	38.2
	Non-vegetarian	1209	61.8
Age	16–25	707	36.1
	26–35	783	40.0
	36–45	360	18.4
	46–55	85	4.3
	56–65	16	0.8
	66–71	5	0.3

**Table 2 ijerph-18-12285-t002:** Prevalence of sleeping disorders.

Sleeping Disorder	Variables	Total *n* (%)	Male *n* (%)	Female *n* (%)
Insomnia ^1^	Yes	961 (49.1%)	133 (43.3%)	828 (50.2%)
	No	995 (50.9%)	174 (56.7%)	821 (49.8%)
Sleepiness ^2^	Yes	451 (23.1%)	41 (13.4%)	410 (24.9%)
	No	1505 (76.9%)	266 (86.6%)	1239 (75.1%)

^1^ Insomnia was defined as individuals who scored >= 6 points in AIS. ^2^ Sleepiness was defined as individuals who scored >= 10 points in ESS.

**Table 3 ijerph-18-12285-t003:** Prevalence of sleeping disorders among females and males.

Gender	Sleeping Disorder	Variables	Vegetarians *n* (%)	Non-Vegetarians *n* (%)	*p*-Value
Females	Insomnia ^1^	Yes	311 (46.6%)	517 (52.7%)	0.014
		No	357 (53.4%)	464 (47.3%)	
	Sleepiness ^2^	Yes	142 (21.3%)	268 (27.3%)	0.0052
		No	526 (78.7%)	713 (72.7%)	
Males	Insomnia ^1^	Yes	29 (36.7%)	104 (45.6%)	ns
		No	50 (63.3%)	124 (54.4%)	
	Sleepiness ^2^	Yes	7 (8.9%)	34 (14.9%)	ns
		No	72 (91.1%)	194 (85.1%)	

^1^ Insomnia was defined as individuals who scored >= 6 points in AIS. ^2^ Sleepiness was defined as individuals who scored >= 10 points in ESS.

**Table 4 ijerph-18-12285-t004:** Prevalence of insomnia and sleepiness stratified by diet and lifestyle variables.

Category	Variables	Insomnia ^1^		*p*-Value	Sleepiness ^2^		*p*-Value
		Yes *n* (%)	No *n* (%)		Yes *n* (%)	No *n* (%)	
Vegetarian diet	Yes	340 (45.5%)	407 (54.5%)	0.01	149 (20.0%)	598 (80.0%)	0.01
	No	621 (51.4%)	588 (48.6%)		302 (25.0%)	907 (75.0%)	
Fruit and vegetables per day ^3^	At least 1	828 (48.2%)	888 (51.8%)	0.03756	379 (22.1%)	1337 (77.9%)	0.006
	Less than 1	133 (55.4%)	107 (44.6%)		72 (30.0%)	168 (70.0%)	
Meat per day ^4^	At least once	335 (48.9%)	350 (51.1%)	ns	155 (22.6%)	530 (77.4%)	ns
	Less than once	626 (49.2%)	645 (50.8%)		296 (23.3%)	975 (76.7%)	
Coffee ^5^	Everyday	51 (48.6%)	54 (51.4%)	ns	24 (22.9%)	81 (77.1%)	ns
	Less often	910 (49.2%)	941 (50.8%)		427 (23.1%)	1424 (76.9%)	
Alcohol ^6^	Everyday	14 (51.9%)	13 (48.1%)	ns	5 (18.5%)	22 (81.5%)	ns
	Less often	947 (49.1%)	982 (50.9%)		446 (23.1%)	1483 (76.9%)	
Smoking	Yes	141 (58.0%)	102 (42.0%)	0.003	64 (26.3%)	179 (73.7%)	ns
	No	820 (47.9%)	893 (52.1%)		387 (22.6%)	1326 (77.4%)	
Using electronic devices 1 h before sleep	Yes	873 (50.8%)	847 (49.2%)	0.0001	407 (23.7%)	1313 (76.3%)	ns
	No	88 (37.3%)	148 (62%)		44 (18.6%)	192 (81.4%)	
BMI [kg/m^2^]	<18.5	59 (53.6%)	51 (46.4%)	0.00458	22 (20.0%)	88 (80.0%)	ns
	18.5–24.99	624 (46.7%)	732 (53.3%)		319 (23.2%)	1055 (76.8%)	
	>24.99	260 (55.1%)	212 (44.9%)		110 (23.3%)	362 (76.7%)	

^1^ Insomnia was defined as individuals who scored >= 6 points in AIS. ^2^ Sleepiness was defined as individuals who scored >= 10 points in ESS. ^3^ Fruit and vegetables per day was defined as individuals who consumed at least one portion of fruit or vegetables everyday. ^4^ Meat per day was defined as individuals who consumed at least one portion of meat everyday. ^5^ Coffee was defined as individuals who drunk coffee 5 or less hours before sleep everyday. ^6^ Alcohol was defined as individuals who consumed alcohol everyday.

## Data Availability

The data presented in this study are available on request from the corresponding author. The data are not publicly available due to not obtaining a consent from respondents for publishing data.

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
