# Peer review of "Sleeping Disorders in Healthy Individuals with Different Dietary Patterns and BMI, Questionnaire Assessment"

_ijerph, 2021, doi:10.3390/ijerph182312285_

Round 1

Reviewer 1 Report

Introduction

  1. The statements in introduction are chaotic, and the aims of this study are drifting. The introduction section should be further reorganized to lead to the clear study aims.
  2. What's the necessity to mention COVID-19 in this study?

METHODS

  1. How's the representativeness of this study? Please state the sampling scheme of this study.
  2. How did the two-week dietary intake (from February 5, 2021 to February 23, 2021) reflect the dietary pattern? People have different dietary patterns in the four seasons yearly.
  3. It's seemed odd to include “diet” as “demographic variable”.
  4. Lines 87-90: Please state how to analyze the dietary patterns from the 29 questions? Based on the concept of priori or posterior (such as factor analysis) methods to derive the dietary pattern? If priori method, please state why and how to score?
  5. ”Diet pattern included: vegetarian, vegan, or including meat. Vegetarians were defined as people who exclude meat; vegans were defined as people who exclude all animal products from their diet. Later when referring to “vegetarians” both vegetarians and vegans are included in that group.”

Please cite references or explain whether this classification has been validated?

  1. “For the assessment of dietary habits, Polish questionnaire KomPAN[21] was used. Due to the limitations of the online survey, we used a minimal set of questions (which included 29 questions) indicated by the authors of this questionnaire.”

Lines 98-99: please state how to validate the short-form questionnaire based on KomPAN?

  1. Please assess the data quality in this study. These data were web-based and self-reported.
  2. Did this study review by IRB? Every subject signed the informed consent?
  3. Lines 106-110: Was the cutoff, 6 points, for insomnia, based on reference #22 or other reference? Please cite the correct reference.
  4. Lines 111-112: Was the cutoff, 10 points, for sleepiness, based on reference #23 or other reference? Please cite the correct reference.

Results

  1. Please integrate those results of univariate analysis into one table.
  2. Was the insomnia prevalence rate nearly half reasonable?
  3. Most results were based on univariate analysis, no multivariate analysis result. No confounding effect had been controlled, so this study was hard to draw conclusion.

Reviewer 2 Report

This study examines the relationship between sleep and diet in coronary heart disease, and from what I have read in the summary, it is an interesting study. However, there are some problems in this study.

 First of all, in the summary, it states that "The aim of the study was to assess the prevalence of sleeping disorders during the COVID-19 pandemic among people with different dietary patterns." However, since there is no information before the COVID-19 pandemic, the effect of the COVID-19 cannot be assessed.

The title says, "Sleeping disorders in healthy individuals with different dietary patterns and BMI, questionnaire assessment", and there seems to be some analysis and tables on BMI, but there is no mention of BMI in the abstract.

 In the abstract, it says that "There is more research needed to establish which dietary factors contribute to the lower prevalence of sleeping disorders among vegetarians. vegetarians.” Howewer, This study does not seem to clarify whether dietary factors contribute to sleeping disorders among vegetarians or not. For example, it could be explained by the influence of people who are conscious of choosing vegetarianism, people who are knowledgeable and educated about such matters, or people who are financially secure. It may be necessary to adjust for such factors in the survey.

In the Introduction section, authors mentioned that "Therefore, it is pivotal that health care workers pay attention to this issue when dealing with patients. I thought authors mentions about local residents, but suddenly it changes to patients, which is confusing.

Authors mentioned in the introduction section that“However, available research does not show a clear causality: does poor sleep affect diet choices or unhealthy diet causes poor sleep? Furthermore, there is little research analyzing the association between a vegetarian diet and sleep quality.” However, this study did not clarify the causal relationship.

Method

 The participants were “random people from Poland who were members of social media groups." There is no mention of age, gender, or other conditions, Information about inclusion criteria is needed.

The data collection was conducted via the Internet, we would like to know about the validation of the question about the dietary patterns. In addition, there is a possibility that people may change their diet behaviors, how long is needed to be “vegetarian” in this study. There is no mention about it in the manuscript.

The authors mentioned that “Assessment of factors that may influence sleep quality", I think social economic status is essential as factors that may influence sleep quality. there is no information about SES, such as income, work or education status.

Regarding statistical analysis, authors mentioned that "Descriptive statistics and chi-square test for categorical variables were used for the analysis of the obtained data. The results were considered statistically significant if p value was less than 0.05." However, this information seems to be insufficient. What was how analyzed is needed.

Regarding the table of results, they seemed to be improved with the analysis.

Reviewer 3 Report

This is an interesting study conducted among a group of 1956 respondents who met eligibility criteria for the inclusion. Authors investigated impact of COVID-19 pandemic on sleeping disorders in healthy individuals with different dietary patterns and BMIs including vegan and non-vegan foods in conjunction with questionnaire assessment . They also aimed to search for correlations between levels of burnout and other mental health disorders and to emphasize specific risks by structured questionnaire and validated instruments in Poland. While I am perfectly aware of this study's utmost importance, still some criticism needs to be addressed:

Abstract

This section should be structurized according to MDPI policy - see https://www.mdpi.com/journal/healthcare/instructions

Also adding few more keywords should improve availability through search engines and eventually lead to improved citation odds

Introduction

L41-49 - while there is a very good, brief and concise sleep description authors didn't mention obstructive sleep apnea screening, which is very important and on the rise; please include short information with relevant citation -  https://www.mdpi.com/2076-3417/11/9/3764

L44 - numeration of references should go as [6-10] and these system should be continued throughout the text - please follow MDPI citation policy

L57-L62 - 'Research from Sicily examined associations between the Mediterranean diet and sleep quality. They observed a positive association between sleep quality and vegetables and fruit intake; and a negative association with confectionery and unhealthy snacks consumption[17]. Another study from Japan shows that higher intake of vegetables, mushrooms, potatoes, seaweed, soy, and eggs was associated with a lower risk of experiencing troubles with falling asleep [18]' - these sentences should be moved and incorporated into Discussion section

The description of hypothesis is not very clear, please add a few more words to the description and study expectations.

Materials and methods

L69-L77 - questionaire data should be available via link or put into Supplement section

Was survey was approved by the local ethics committee ? - please provide number and name of the Commitee in full if yes, or write if irrelevant.

Results

Questionnaires used and statistical methods are appropriate and up for the task.

Discussion

This section is slightly too short, discusses 14 references only and reads slightly like an explanation of a results. An entire manuscript would greatly benefit, if authors discussed their findings with more similar papers published recently, concerning healthcare/pandemic issues, both economically and psychologically focused.

L212-214 - while most of the publications emphasize male predominant pattern of sleep-related breathing disroders there is some conflicting evidence in terms of age-related diminishing testosterone levels and OSA - please check and briefly discuss e.g. https://pubmed.ncbi.nlm.nih.gov/29774669/

L241-246 - as authors wrote that tryptophan is precursor for melatonin and serotonin, it could be valid to mention studies concerning role of serotonin and melatonin in sleep disorders, as there is none included - with relevant citation(s)

Round 2

Reviewer 1 Report

Intruduction

Point 1: The statements in introduction are chaotic, and the aims of this study are drifting. The introduction section should be further reorganized to lead to the clear study aims.

Response 1: We reorganised the introduction slightly, and expanded on the issue of the study aims and study hypothesis.

Comment (round 2): No more comment

Point 2: What's the necessity to mention COVID-19 in this study?

Response 2: The study does not intend to examine the influence of COVID-19 pandemic on sleep quality. However, we believe it is important to highlight the fact that the data were collected during that time, as it could have interfered with the results. To the introduction section, I added a paragraph which explains that.

Comment (round 2): No more comment

Methods

Point 1: How's the representativeness of this study? Please state the sampling scheme of this study.

Response 1: Our questionnaire was distributed via Facebook groups and profiles. We shared our study mostly via groups which gathered vegetarians, hence the high rate of vegetarians in our study. Another type of groups in which we shared our survey were student’s group. However, most of our respondents come from Damian Parol’s facebook fanpage, which is liked by 63 174 people – this enabled us to reach more varied group of people. At the same time, we are aware that there can be some bias among our group, due to the fact that both vegetarian groups and Damian Parol’s profile (who is a dietitian) may gather mostly people who are interested in healthy lifestyle.

Comment (round 2): The study subjects obtained by FB fanpage and student’s group have no representativeness. This is a major flawing of research method.

Point 2: How did the two-week dietary intake (from February 5, 2021 to February 23, 2021) reflect the dietary pattern? People have different dietary patterns in the four seasons yearly.

Response 2: In the KomPAN questionnaire which we used for assessing the diet, respondents are asked to report how often they consumed certain products during the last year, not only during the last two weeks. They could choose one of six options: never, 1-3 times per month, once a week, few times a week, once a day, few times a day.  I added that explanation to the method section.

Comment (round 2): No more comment

Point 3: It's seemed odd to include “diet” as “demographic variable”.

Response 3: Thank you for this comment. We changed the title of this paragraph to “demographic information and declaration of diet”.

Comment (round 2): No more comment

Point 4: Lines 87-90: Please state how to analyze the dietary patterns from the 29 questions? Based on the concept of priori or posterior (such as factor analysis) methods to derive the dietary pattern? If priori method, please state why and how to score?

Response 4: As I mentioned in point 2, respondents were asked to report how frequently they consumed certain products during the last year. Therefore, it was based on the concept of posteriori. Scoring protocol of this questionnaire is meticulously described in the reference #24. What is more, main dietary patterns indicated in our study are: vegetarian and non-vegetarian. We used the results of the KomPAN questionnaire to assess how often respondents consumed certain groups of products such as: fruit and vegetables, meat, coffee or alcohol.

Comment (round 2): No more comment

Point 5:”Diet pattern included: vegetarian, vegan, or including meat. Vegetarians were defined as people who exclude meat; vegans were defined as people who exclude all animal products from their diet. Later when referring to “vegetarians” both vegetarians and vegans are included in that group.”

Please cite references or explain whether this classification has been validated?

Response 5: I added proper citation and specified our classification.

Comment (round 2): No more comment

Point 6: “For the assessment of dietary habits, Polish questionnaire KomPAN[21] was used. Due to the limitations of the online survey, we used a minimal set of questions (which included 29 questions) indicated by the authors of this questionnaire.”

Lines 98-99: please state how to validate the short-form questionnaire based on KomPAN?

Response 6: As we mentioned, this short form of the KomPAN questionnaire is described by the authors of it in the reference #24. There, you can find the scoring protocol for both whole KomPAN questionnaire as well as for the short form. The short form of questionnaire was validated by its authors.

Comment (round 2):  Please cite references to show the validation information of the short-form KomPAN questionnaire.

Point 7: Please assess the data quality in this study. These data were web-based and self-reported.

Response 7: Our questionnaire was web-based and self-reported. Therefore, we cannot be sure that respondents fully understood the questions and answered them correctly. What is more, due to the socio-cultural pressure to follow a “correct” dietary pattern, respondents whose intake of supposedly healthy food is low may be tempted to report their intake of these products as higher than it really is. Furthermore, the intake of unhealthy food may be underreported.

However, to raise the quality of the study, some questionnaires were excluded from analysis due to obtaining unattainable answers in which a high probability of untruth was predicted (e.g., BMI = 5 kg/m2). What is more, due to the scoring protocol of the KomPAN questionnaire, we excluded a few questionnaires which were untrustworthy due to the contradictions among answers. We mention that in the methods section.

Comment (round 2): The authors must state that the questionnaires are designed for online surveys or show the reliability and validity of the questionnaires for online using.

Point 8: Did this study review by IRB? Every subject signed the informed consent?

Response 8: According to the policies of our University, this kind of study (questionnaire based) did not need to be reviewed by IRB. A statement about the aims of the study and the declaration of anonymity and confidentiality were included in the form. Filling in the questionnaire and clicking the „send” button was synonymous with informed consent to participate in the study.

Comment (round 2): No more comment

Point 9: Lines 106-110: Was the cutoff, 6 points, for insomnia, based on reference #22 or other reference? Please cite the correct reference.

Response 9: Yes, the cut off for insomnia was based on the mentioned reference #22 (in the updated version of manuscript number of this reference is 25). I added proper citation.

Comment (round 2): No more comment

Point 10: Lines 111-112: Was the cutoff, 10 points, for sleepiness, based on reference #23 or other reference? Please cite the correct reference.

Response 10: Yes, the cut off for sleepiness was based on the reference #23 (in the updated version of manuscript number of this reference is 26). I added proper citation.

Comment (round 2): No more comment

Results

Point 1: Please integrate those results of univariate analysis into one table.

Response 1: Do I understand correctly, that what You ask for is to show all the study results in one table?

Comment (round 2):

  1. The numbers of men and women and the numbers of vegetarians and non-vegetarians can be found in table 3 and table 4. No need to show table 1 and table 2.
  2. The age group information in table 1 can be described in texts.
  3. Table 3 and table 4 can be integrated into one
  4. Information in table 7 can be integrated into table 5 and table 6, respectively.
  5. And, table 5 and table 6 can be integrated into one.

Point 2: Was the insomnia prevalence rate nearly half reasonable?

Response 2: We are fully cognizant of the fact that this is very high rate of insomnia. However, a few studies which we cited had observed similar insomnia rate. There was another study conducted in Poland in very similar time as our study, and it draws a similar conclusion. The PaLS study indicated a 58,13% insomnia rate among students. In a study conudcted in Italy (Cellini et. al.) insomnia rate reached 52,4%. We mention this study in the discussion section.

Comment (round 2): No more comment

Point 3: Most results were based on univariate analysis, no multivariate analysis result. No confounding effect had been controlled, so this study was hard to draw conclusion.

Response 3: We are aware of that and included it into the limitations section.

Comment (round 2): This study can perform multivariate analysis, instead of writing this issue in limitation section only.

Reviewer 3 Report

All of my former remarks were successfully addressed. Some very minor issues yet still persist:

L68-77 - hypothesis is almost illegible, please clarify. The sentence 'Other study[20] shows that higher meat consumption was correlated with poor sleep quality, and decrease in sleep duration' should be moved on top of the paragraph. Please check any other peer-reviewed articles for examples, how the hypothesis should be formed. Also sentence about Covid: 'This study does not intend to examine the influence of COVID-19 pandemic on sleep quality. However, it is important to highlight the fact that the study was conducted during COVID-19 pandemic, as it could have interfered with the results' is vague and should be removed or put into appropriate paragraph - study limitations.

L138 - Statistica software manufacturer proper data is missing - Tibco software HQ is located in Palo Alto, CA, US - please update accordingly

L142 - 'borderline' is vague - please paraphrase the sentence or remove redundant word.

L291 - cigarette use - please try to avoid repetitions as 'smokers' is used three times in a row.

L319-320 - 'The question is whether short sleep duration causes a change in BMI or a higher BMI results in a change in sleep duration?' - this question should be moved into Conclusions section and incorporated into further directions.

References

Please carfully check all the included references style with MDPI publishing policy. Also:

L365 - Reference 1 - add source (website) and access date

L420 - Reference 24 - provide English version if identical with native, if not it should be put into Suplement section. Also source and access date should be included.
